# Postoperative complication management: How do large language models measure up to human expertise?

Sophie-Caroline Schwarzkopf[1,2☯], Jean-Paul Bereuter[1☯], Mark Enrik Geissler[1☯], Jürgen Weitz[1,3], Marius Distler[1], Fiona R. Kolbinger [1,4,5,6]*

1 Department of Visceral, Thoracic and Vascular Surgery, University Hospital and Faculty of Medicine Carl Gustav Carus, TUD Dresden University of Technology, Dresden, Germany, 2 Department of Obstetrics and Gynecology, University Hospital and Faculty of Medicine Carl Gustav Carus, TUD Dresden University of Technology, Dresden, Germany, 3 Centre or Tactile Internet with Human-in-the-Loop (CeTI), TUD Dresden University of Technology, Dresden, Germany, 4 Weldon School of Biomedical Engineering, Purdue University, West Lafayette, Indiana, United States of America, 5 Regenstrief Center for Healthcare Engineering (RCHE), Purdue University, West Lafayette, Indiana, United States of America, 6 Department of Biostatistics and Health Data Science, Richard M. Fairbanks School of Public Health, Indiana University, Indianapolis, Indiana, United States of America

☯ These authors contributed equally to this work.
* fiona.kolbinger@uniklinikum-dresden.de

## Abstract

Managing postoperative complications is an essential part of surgical care and largely depends on the medical team's experience. Large Language Models (LLMs) have demonstrated immense potential in supporting medical professionals. To evaluate the potential of LLMs in surgical patient care, we compared the performance of three state-of-the-art LLMs in managing postoperative complications to that of a panel of medical professionals based on six postsurgical patient cases. Six realistic postoperative patient cases were queried using GPT-3, GPT-4, and Gemini-Advanced and presented to human surgical caregivers. Humans and LLMs provided a triage assessment, an initial suspected diagnosis, and an acute management plan, including initial diagnostic and therapeutic measures. Responses were compared based on medical contextual correctness, coherence, and completeness. In comparison to human caregivers, GPT-3 and GPT-4 possess considerable competencies in correctly identifying postoperative complications (humans: 76.3% vs. GPT-3: 75.0% vs. GPT-4: 96.7%, p = 0.47) as well as triaging patients accordingly (humans: 84.8% vs. GPT-3: 50% vs. GPT-4: 38.3%, p = 0.19). With regard to diagnostic and therapeutic management of postoperative complications, GPT-3 and GPT-4 provided comprehensive management plans. Gemini-Advanced often provided no diagnostic or therapeutic recommendations and censored its outputs. In summary, LLMs can accurately interpret postoperative care scenarios and provide comprehensive management recommendations. These results showcase the improvements in LLMs performance with regard to postoperative surgical use cases and provide evidence for their potential value to support and augment surgical routine care.

**Data availability statement:** Detailed case descriptions and ideal solutions are provided in Supplementary Table 1. Participant data analyzed in the context of this work are publicly available at https://games.jmir.org/2023/1/e44708/.

**Funding:** The author(s) received no specific funding for this work.

**Competing interests:** The authors have no conflicts of interest to declare.

**Abbreviations:** AI, Artificial Intelligence; COVID-19, Coronavirus disease; GPT-3, Generative Pre-trained Transformer 3; GPT-4, Generative Pre-trained Transformer 4; LLM, Large language models; POPF, Postoperative pancreatic fistula; RAG, Retrieval-augmented generation.

## Introduction

Accurate identification and timely treatment of postoperative complications can greatly reduce their impact on patients. Therefore, situational awareness and decision-making competencies are crucial in surgical routine care. [1] Large language models (LLMs) have demonstrated potential in the context of medical decision support. [2,3] However, evidence concerning their usability in surgery remains limited. [4] This study aimed to assess the validity of three publicly accessible LLMs in surgical routine care by examining representative use cases related to the management of postoperative complications. Specifically, we compared the output of three LLMs, Generative Pre-trained Transformer 3 (GPT-3), GPT-4, and Gemini-Advanced, with guideline-based clinical gold standards and compared LLM performance to that of a human caregiver cohort in the context of postoperative complication management.

LLMs, such as GPT-3, GPT-4 and Gemini, are trained on large datasets and utilize a deep neural network architecture to predict and generate human-like text. [5,6] Therefore, they can be used for various applications, including automated question answering, text summarization, and content generation. [7,8] The most recent model, GPT-4, is capable of additional tasks such as image and audio processing. [9]

Based on these capabilities, LLMs have shown promising results in supporting medical professionals in basic tasks. [10] For instance, they have are capable of answering medical licensing and board exam questions, [11–13] summarizing and structuring clinical reports [14,15] and enhancing preliminary clinical diagnoses. [16] Beyond basic tasks, LLMs have proven valuable in supporting clinicians with complex tasks such as clinical decision making regarding differential diagnoses, diagnostic testing, final diagnosis, and therapeutic management. [3,17,18]

This assistance could be particularly valuable in the context of surgery for managing postoperative complications, a critical aspect of surgical care requiring prompt recognition, accurate diagnosis, and appropriate intervention. [19,20] Since LLMs can retrieve and summarize relevant information and analyze these data in the context of vast amounts of medical literature, clinical guidelines, and patterns in patient data, they are well-suited to provide evidence-based recommendations for managing postoperative complications. [21] This could help caregivers diagnose and treat patients more accurately and alert them to potential problems or complications that may arise in the future.

However, the exact potential of LLMs in patient care, particularly in surgery and postoperative complication management, remains to be elucidated, particularly in the light of the continuous evolution of currently available LLMs. As LLMs evolve and adapt to new data and feedback, a phenomenon known as 'behavior drift' may occur. This refers to the unexpected changes in an LLM's behavior as it learns from additional input and feedback. While this behavior drift presents challenges, it may also enhance the clinical applicability of LLMs. [22]

Understanding the clinical implications of LLMs in surgery and postoperative complication management is the central aim of our study. Here, we empirically examine

the potential of LLMs in surgical patient care and compare their competencies in managing postoperative complications to those of a cohort of human caregivers. By considering three different LLMs, GPT-3, GPT-4, and Gemini, we showcase the distinct capabilities and sources of error that occur across generations of LLMs, illustrating both their translational potential and the respective caveats in surgical patient care.

## Methods

### Conceptualization of patient cases

We conceptualized six independent and clinically relevant postoperative patient cases, as described previously. [23] These cases represent common complications following pancreatic surgery (such as postpancreatectomy hemorrhage and postoperative pancreatic fistula [POPF]), colorectal surgery (including anastomotic leakage and mechanical ileus), and general postoperative complications (e.g., stroke, COVID-19/wound infection). The used cases were developed and illustrated utilizing anonymized images, data, and histories of postoperative patients. The simulated patients were two to a maximum of 14 days post their primary surgery. The initial case description gave limited information about the patient, as would be realistic in a clinical management scenario for a previously unseen patient, prior to initiating appropriate diagnostics, such as laboratory or imaging examinations, and starting corresponding therapeutic measures. [23] Ideal case and complication management pathways were based on European guidelines and scientific evidence. [24–27] Detailed case descriptions and ideal solutions are provided in S1 Table.

### Case evaluation – LLMs

The six postoperative patient cases were evaluated by three different LLMs: GPT-3 (released in June 2020), GPT-4 (released in March 2023), and Gemini 1.0 Pro (released in December 2023). Testing was performed between February 1 and March 31, 2024 for GPT-4 and Gemini and in December 2022 for GPT-3. Additional testing using GPT-4o was performed between April 15th and 17th, 2025. The additional testing included counterfactual modifications to age, postoperative day and blood pressure in the anastomotic leakage patient case, as well as temperature adjustments to the model. Prompting was performed without additional augmented retrieval generation (RAG) or in context learning to test a zero-shot approach. Therefore, models were only tested upon their baseline training data. Each LLM analyzed each case using a three-step testing process:

1. **Triage**: First, LLMs were prompted to determine if the case represents an emergency. An emergency was defined as a sudden event posing an immediate threat to the patient's life and health, characterized by threatened, disturbed, or failed vital functions due to injury or acute illness. [28] Based on this definition, stroke and POPF were classified as emergencies, whereas sentinel bleeding, anastomotic leakage, mechanical ileus and COVID-19/wound infection were not considered emergencies.

   Prompt: "Is this patient case an emergency?"

2. **Identification of the complication**: This included the identification of the organ system involved (cardiac, respiratory, neurologic, gastrointestinal, renal, or other organ system), the mechanism of the complication (infection, bleeding, or other complication mechanism), and the specific complication.

   Prompt: "Which spectrum do you suspect the postoperative complication to be in? Options are: Cardiac spectrum, Respiratory spectrum, Neurologic spectrum, Gastrointestinal spectrum, Renal spectrum, Other spectrum.
   You may optionally specify the suspected diagnosis. Please select between: Infection complication, Bleeding complication, Other complication."
   Furthermore, we tested LLMs' diagnostic potential directly by using the prompt "What is the initial suspected diagnosis?".

3. **Description of appropriate management**: Last, LLMs were prompted to outline the diagnostic and therapeutic measures to manage the identified complication.

Prompt: "What diagnostic and therapeutic measures would you order as part of the acute management?"

For the evaluation of triage and the initial suspected diagnosis, each case was processed ten times by each tested LLM to evaluate the variability of LLM outputs. For the evaluation of initial management, each case was processed once by each LLM.

## Case evaluation – Human caregivers

Human management was evaluated as described previously. [23] In brief, human caregivers with varying experience in abdominal surgery independently assessed the realistic patient cases. A total of 131 cases were assessed. The cohort of human caregivers reflected the whole spectrum of healthcare professionals and consisted mainly of medical students (74/131, 56%), general surgery residents and board-certified surgeons (19/131, 15%), and surgical experts (senior and chief physicians) (18/131, 14%). In addition to these participants, nursing staff and other clinical staff with patient contact were included (20/131, 15%). The participants were recruited from the entire German-speaking region, particularly from the medical and nursing staff of the Carl Gustav Carus University Hospital and the students of the Carl Gustav Carus Medical Faculty in Dresden, Germany. Following a brief case presentation, participants were asked to triage the virtual patient and make an initial suspected diagnosis. The distribution of cases was as follows: anastomotic leakage (n = 60), stroke (n = 19), POPF (n = 14), mechanical ileus (n = 10), COVID-19/ wound infection (n = 10), and sentinel bleeding (n = 18). As a limitation of the underlying human case evaluation study, no specific sample size considerations or power analyses were conducted. In addition, participants selected cases using a drop-down function, where anastomotic insufficiency was the first case and therefore selected by most participants. S2 Table displays the diagnostic performance of human caregivers of various professional levels in relation to the affected organ system, the complication mechanism, and the exact complication, as published previously. [23]

## Comparative analysis of the case evaluation

Responses from human caregivers and LLMs were comparatively analyzed. The comparative analysis comprised two main components: the emergency assessment of the patient (triage) and the identification of the specific underlying complication (Fig 1). In addition, we analyzed to what extent the acute management pathways suggested by GPT-3, GPT-4, and Gemini comprised redundant, dangerous, inconsistent, or useless measures or whether important measures had been missed. LLM output regarding the complication management was assessed by two independent raters (physicians with three and five years of expertise in surgical routine care).

## Statistics and visualization

Descriptive statistics were conducted using GraphPad Prism version 9. Visualizations were created using GraphPad Prism version 9 and Canva. The given values are displayed as distributions of frequencies for discrete variables. Depending on the data characteristics, the proper statistical test was utilized (ordinary one-way ANOVA with multiple comparisons) to perform between-group comparisons. P-values below 0.05 were considered statistically significant.

## Results

### Emergency assessment of postoperative cases

We compared humans' and LLMs' assessments of whether or not the presented postoperative cases present an emergency. LLMs displayed heterogeneous responses and performances when classifying the urgency of postoperative cases (Table 1).

| Patient cases | Case 1 Anastomotic leakage | Case 2 Stroke | Case 3 POPF | Case 4 Mechanical ileus | Case 5 COVID-19 / Wound infection | Case 6 Sentinel bleeding |
|---|---|---|---|---|---|---|
| Data (gender / age in years / postoperative day (POD) / surgery) | M / 62 / POD: 4 / Stoma reversal | M / 83 / POD: 4 / Right hemicolectomy | F / 65 / POD: 4 / Distal pancreatectomy | M / 87 / POD: 2 / Left hemicolectomy with protective ileostomy | F / 88 / POD: 14 / Total rectal extirpation | F / 71 / POD: 7 / Whipple |
| Current Symptoms | Stool is noticed in the drainage. Abdomen: bowel sounds in all four quadrants, guarding in left upper/lower abdomen, skin is unremarkable. No heart murmurs. Regular breath sounds. | Patient is no longer speaking and is dazed. Nurse has checked and secured vital signs, i.v. access is in place. It is unclear what mental status the patient came in. | Patient presents in a worsened general condition with guarding in the upper abdomen, and states abdominal pain radiating to the back. Blood pressure: 105/70, pulse slightly elevated. Unremarkable surgical scar. Renal positional palpitation appears to be present. No indwelling catheter. Temp.: 39.4°C. | Patient presents today with abdominal pain and nausea. An image of the abdomen is sent to you: abdomen is distended, stoma bag is empty. The patient has already started to build up his diet. The surgical scar looks good, drainage fluid is clear. | Family physician re-admits patient with rising CRP. Wound looks infected. Other symptoms include fever, headache, and circulatory problems. A COVID-19 rapid test was done in the emergency room and negative. PCR result is pending. No heart murmurs. Basal attenuated breath sounds. | Drainage was already tinged with blood over the weekend. Today (monday) the drainage fluid appears bloody again. Deteriorated general condition, circulatory dysfunction. The surgical scar looks good, the patient expresses diffuse abdominal pain. |

All cases were presented to GPT-3, GPT-4, Gemini-Advanced and Human caregivers and asked for an assessment in relation to triage, first suspected diagnosis and initial management. The answers were evaluated in relation to medical guidelines.

**Fig 1. Overview of the study design.** Six postoperative patient cases (anastomotic leakage, stroke, postoperative pancreatic fistula (POPF), mechanical ileus, COVID-19/wound infection, sentinel bleeding) were presented to human caregivers and three different LLMs (GPT-3, GPT-4 and Gemini). Responses were compared and analyzed regarding alignment with medical guidelines. Abbreviations: Postoperative day (POD); Postoperative Pancreatic Fistula (POPF).

**Table 1. Triage of the six postoperative cases by LLMs and human caregivers.** Triage (i.e., assessment if a case presents an emergency) was conducted by LLMs (GPT-3, GPT-4, Gemini) and human caregivers. The table outlines the number of correct triage assessments. Gemini censored all outputs.

| Case | Stroke | POPF | Sentinel bleeding | Anastomotic leakage | Mechanical ileus | COVID-19/ wound infection |
|---|---|---|---|---|---|---|
| Correct classification | Emergency | Emergency | No Emergency | No Emergency | No Emergency | No Emergency |
| Gemini | NA | NA | NA | NA | NA | NA |
| GPT-4 | 10/10 (100%) | 10/10 (100%) | 0/10 (0%) | 0/10 (0%) | 0/10 (0%) | 3/10 (30%) |
| GPT-3 | 10/10 (100%) | 10/10 (100%) | 0/10 (0%) | 0/10 (0%) | 0/10 (0%) | 10/10 (100%) |
| Humans (overall) | 18/19 (95%) | 13/14 (93%) | 1/18 (6%) | 2/60 (1%) | 2/10 (20%) | 5/10 (50%) |
| Experts | 3/3 (100%) | 2/2 (100%) | NA | 0/11 (0%) | 0/1 (0%) | 1/1 (100%) |
| Board-certified surgeons | 1/1 (100%) | 1/1 (100%) | 0/1 (0%) | 0/2 (0%) | NA | NA |
| Surgical Residents | 1/1 (100%) | 2/2 (100%) | 0/4 (0%) | 0/4 (0%) | 0/1 (0%) | 0/2 (0%) |
| Medical students (years 1–5) | 4/5 (80%) | 2/2 (100%) | 1/3 (66.7%) | 0/15 (0%) | 1/3 (33.3%) | 1/2 (50%) |
| Medical students (final year, year 6) | 7/7 (100%) | 4/5 (100%) | 0/8 (0%) | 1/17 (5,9%) | 0/4 (0%) | 2/3 (66.7%) |
| Non-medical staff | 2/2 (100%) | 2/2 (100%) | 0/2 (0%) | 1/11 (9%) | 1/1 (100%) | 1/2 (50%) |

Gemini did not output any triage assessments, consistently stating its status as a language model and its inability to process and respond to medical cases. In contrast, GPT-3 correctly assessed the stroke and POPF cases as emergencies and correctly classified the COVID-19/wound infection case as no emergency. Nevertheless, GPT-3 overestimated the urgency of the sentinel bleeding, the anastomotic leakage and the mechanical ileus case by misclassifying these cases as an emergency. In instances of positive emergency assessment, GPT-3 provided meaningful reasoning, referencing the initial case presentation.

Similarly, GPT-4 delivered meaningful responses for positive emergency assessments and was able to classify the stroke and POPF as emergencies. However, GPT-4 also overestimated some postoperative cases. Similarly to GPT-3, the urgency of the sentinel bleeding, the anastomotic leakage and the mechanical ileus case was overestimated by misclassifying these cases as an emergency. In addition, the COVID-19/ wound infection case was partly misclassified as an emergency by GPT-4. Specifically, this means that in three instances, GPT-4 referred to the case as an emergency, even though the case was initially classified as not being an emergency.

Difficulties in emergency classification could also be recognized in the human cohort. Here, only the cases of anastomotic insufficiency and postoperative pancreatic fistula were mostly correctly classified. The triage performances between GPT-3, GPT-4 and humans were not significantly different (humans: 84.4% correct vs. GPT-3: 50.0% correct; humans: 84.4% correct vs. GPT-4: 38.3%, p = 0.19) (Fig 2A).

## Identification of the postoperative complication

We next compared the performance of LLMs and human caregivers in identifying the affected organ system, underlying complication mechanism, and exact complication across the six postoperative scenarios.

With regard to identification of the organ system affected by the postoperative complication, all LLMs and most human caregivers correctly rated most of the presented cases (Table 2). Analyzing the performance of LLMs, we observed differences concerning the COVID-19/wound infection case (GPT-3: 0/10 correctly rated vs. GPT-4 10/10 correctly rated). In addition, Gemini demonstrated inferior performance across almost all cases compared to the other two LLMs (Fig 2B). In comparison to human caregivers, LLMs were not significantly different in identifying the affected organ system (humans: 85.7% correctly rated vs. GPT-3: 83.3% correctly rated vs. GPT-4: 100% correctly rated vs. Gemini: 61.7% correctly rated, p = 0.22). The pathophysiological mechanism underlying the complication was correctly identified by the majority of human raters and most LLMs (Fig 2C). GPT-3 and GPT-4 correctly identified the complication mechanism in almost all cases, outperforming Gemini. In comparison to human caregivers, we observed significant performance differences in the identification of the pathophysiological complication mechanism for Gemini (humans: 89.8% correctly rated vs. Gemini: 51.7%, p < 0.01). GPT-3 and GPT-4 did not perform significantly differently to human caregivers (humans: 89.8% correctly rated vs. GPT-3: 97.7% correctly rated, p = 0.29; humans: 89.8% correctly rated vs. GPT-4: 100% correctly rated, p = 0.11).

Last, we analyzed the correctness of LLMs' and humans' suspected complications in all six cases. Here, LLMs showed a similar pattern (Fig 2D). While GPT-4 performed best in rating the suspected complication, GPT-3 and Gemini showed a worse rating, especially for the POPF (GPT-3: 0/10 correctly rated vs. GPT-4: 10/10 correctly rated vs. Gemini: 0/10 correctly rated) and the sentinel bleeding case (GPT-3: 5/10 correctly rated vs. GPT-4: 10/10 correctly rated vs. Gemini: 4/10 correctly rated). Overall, we observed no significant differences between humans and LLMs in correctly identifying the exact complication (humans: 76.3% correctly rated vs. GPT-3: 75.0% correctly rated vs. GPT-4: 96.7% correctly rated, Gemini: 61.7% p = 0.39).

With regard to the identification of the exact complication, we conducted a disaggregated analysis of the performance of the human caregiver subgroups across all six postoperative complication cases (Fig 3, S2 Table). We observed heterogeneous classification patterns among the various healthcare professional subgroups, with performance improvements with increasing professional experience. GPT-4 generally demonstrated performance comparable to or surpassing that of medical students. Surgeons at all experience levels consistently outperformed GPT-4 in correctly identifying the suspected diagnosis. Both GPT-3 and Gemini depicted performance levels that were, on average, inferior to those of medical professionals of all levels.

## Diagnostic and treatment measures for postoperative complications

We evaluated the diagnostic and therapeutic measures proposed by LLMs with regard to redundancy, clinically dangerous potential, inconsistency, and completeness across different cases (Table 3). Gemini refused output related to medical

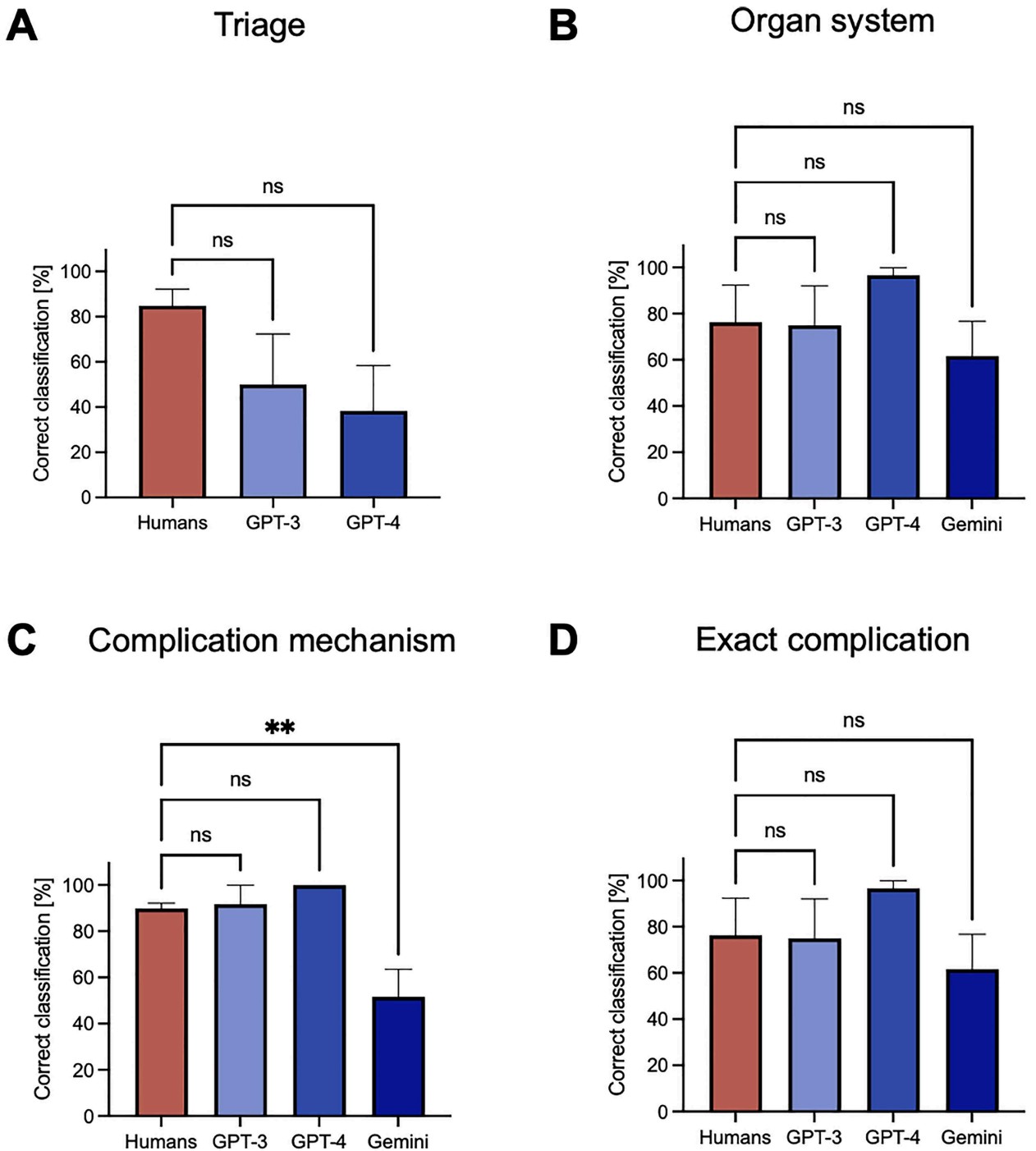

**Fig 2. Emergency assessment and identification of the underlying complication. A:** Emergency assessment was conducted by LLMs (GPT-3, GPT-4, Gemini censored all outputs and was therefore not included in this analysis) and human caregivers. **B:** Identification of the organ system affected by the postoperative complication by LLMs (GPT-3, GPT-4, Gemini) and human caregivers. **C:** Identification of the complication mechanism underlying the postoperative complication by LLMs (GPT-3, GPT-4, Gemini) and human caregivers. **D:** Identification of the exact postoperative complication by LLMs (GPT-3, GPT-4, Gemini) and human caregivers. Statistical analysis was conducted using an ordinary one-way ANOVA. *P<0.05, **P<0.01, ***P<0.001. All data are presented as means and SEM.

**Table 2. Evaluation of the suspected diagnosis by suspected affected organ system, suspected complication mechanism, and suspected exact complication across all postoperative cases by LLMs (GPT-3, GPT-4, Gemini-Advanced) and human caregivers.** The table outlines the number of correct triage assessments. Specification of the organ system was mandatory, while specification of complication mechanism and exact complication were voluntary for human caregivers.

| | Anastomotic leakage | | | | Stroke | | | |
| | LLM | | | Humans | LLM | | | Humans |
| | GPT-3 | GPT-4 | Gemini | | GPT-3 | GPT-4 | Gemini | |
|---|---|---|---|---|---|---|---|---|
| **Organ system** | 10/10 (100%) | 10/10 (100%) | 9/10 (90%) | 59/60 (98%) | 10/10 (100%) | 10/10 (100%) | 9/10 (90%) | 19/19 (100%) |
| **Complication mechanism** | 10/10 (100%) | 10/10 (100%) | 7/10 (70%) | 34/40 (85%) | 10/10 (100%) | 10/10 (100%) | 5/10 (50%) | 11/13 (85%) |
| **Exact complication** | 10/10 (100%) | 10/10 (100%) | 8/10 (100%) | 10/11 (91%) | 10/10 (100% | 10/10 (100%) | 9/10 (90%) | 11/11 (100%) |
| | **POPF** | | | | **Mechanical ileus** | | | |
| | LLM | | | Humans | LLM | | | Humans |
| | GPT-3 | GPT-4 | Gemini | | GPT-3 | GPT-4 | Gemini | |
| **Organ system** | 10/10 (100%) | 10/10 (100%) | 7/10 (70%) | 12/14 (86%) | 10/10 (100%) | 10/10 (100%) | 8/10 (80%) | 10/10 (100%) |
| **Complication mechanism** | 10/10 (100%) | 10/10 (100%) | 3/10 (30%) | 8/9 (89%) | 10/10 (100%) | 10/10 (100%) | 2/10 (20%) | 7/8 (88%) |
| **Exact complication** | 0/10 (0%) | 8/10 (80%) | 0/10 (0%) | 0/2 (0%) | 10/10 (100%) | 10/10 (100%) | 6/10 (60%) | 5/5 (100%) |
| | **COVID-19/ wound infection** | | | | **Sentinel bleeding** | | | |
| | LLM | | | Humans | LLM | | | Humans |
| | GPT-3 | GPT-4 | Gemini | | GPT-3 | GPT-4 | Gemini | |
| **Organ system** | 0/10 (0%) | 10/10 (100%) | 0/10 (0%) | 3/10 (30%) | 10/10 (100%) | 10/10 (100%) | 4/10 (40%) | 18/18 (100%) |
| **Complication mechanism** | 10/10 (100%) | 10/10 (100%) | 10/10 (100%) | 5/5 (100%) | 5/10 (50%) | 10/10 (100%) | 4/10 (40%) | 12/13 (92%) |
| **Exact complication** | 10/10 (100%) | 10/10 (100%) | 10/10 (100%) | 2/3 (67%) | 5/10 (50%) | 10/10 (100%) | 4/10 (40%) | 1/1 (100%) |

cases at all. GPT-4 suggested less redundant diagnostic and therapeutic measures compared to GPT-3 (GPT-3: 3/6 cases vs. GPT-4: 0/6 cases). Similarly, GPT-4 suggested less dangerous (GPT-3: 2/6 cases vs. GPT-4: 0/6 cases) and nonsensical (GPT-3: 4/6 cases vs. GPT-4: 0/6 cases) measures. With regard to the presentation of inconsistent measures, both GPT-3 and GPT-4 demonstrated a low number of inconsistent measures (GPT-3: 1/6 cases vs. GPT-4: 0/6 cases). However, the absence of important measures was more frequent in GPT-3 than GPT-4 (GPT-3: 4/6 cases vs. GPT-4: 2/6 cases). Overall, both GPT-3 and GPT-4 offered meaningful suggestions, but GPT-4 output fewer redundant and dangerous recommendations.

Last, we conducted a preliminary evaluation of counterfactual changes to patient parameters and LLM temperature for the anastomotic leakage case. Here, changes in age, POD, blood pressure, and LLM temperature did not alter the performance of GPT-4o (S3 Table). With regard to the proposed treatment pathway, we observed slight performance variations of GPT-4o with regard to the output of dangerous, inconsistent, or nonsensical measures, or the absence of important measures, yet without a clear pattern related to age, postoperative day, blood pressure, or LLM temperature (S4 Table).

## Discussion

The objective of our study was to empirically examine the potential of LLMs in surgical patient care and compare their competencies in identifying and managing postoperative complications to those of humans. By considering three different

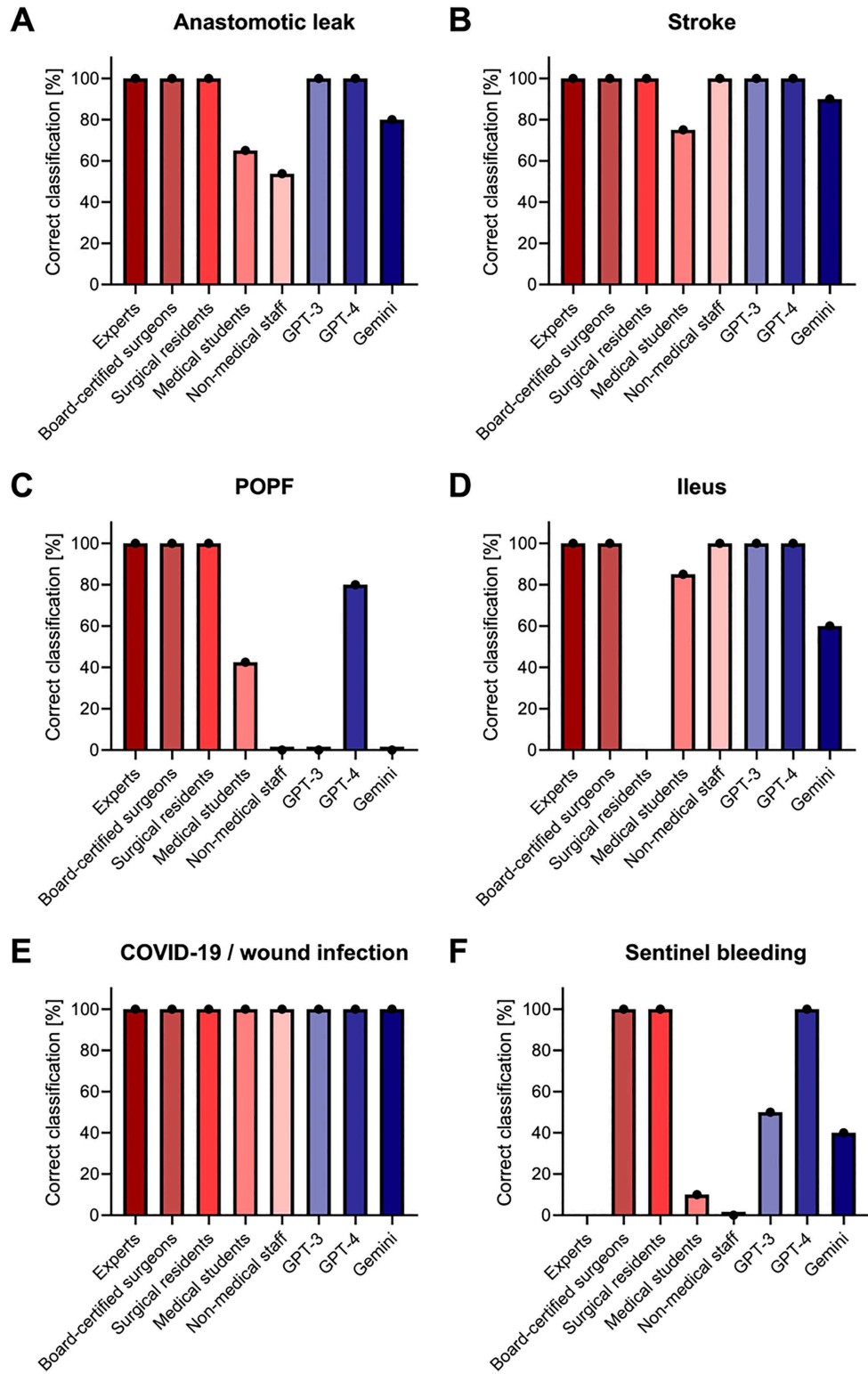

**Fig 3. Disaggregated analysis of the performance of the human caregiver subgroups compared to LLMs concerning the identification of the suspected diagnosis.** The analysis was conducted individually for each of the postoperative complication cases.

Table 3. **Suggested diagnostic and therapeutic measures for the six cases.** The diagnostic and therapeutic pathways proposed by LLMs (GPT-3, GPT-4, Gemini) were evaluated for presence of redundant, dangerous, inconsistent, incomplete, and nonsensical measures. Ticks indicate presence of redundant, dangerous, inconsistent, missing, or nonsensical measures. Crossmarks indicate no presence of the respective characteristics. Gemini censored all outputs.

| Case | Anastomotic leakage | | | Stroke | | | POPF | | |
|---|---|---|---|---|---|---|---|---|---|
| | GPT-3 | GPT-4 | Gemini | GPT-3 | GPT-4 | Gemini | GPT-3 | GPT-4 | Gemini |
| Redundant measures | ✓ | ✗ | NA | ✓ | ✗ | NA | ✗ | ✗ | NA |
| Dangerous measures | ✗ | ✗ | NA | ✓ | ✗ | NA | ✗ | ✗ | NA |
| Inconsistent measures | ✗ | ✗ | NA | ✗ | ✗ | NA | ✓ | ✗ | NA |
| Absence of important measures | ✓ | ✗ | NA | ✓ | ✗ | NA | ✗ | ✗ | NA |
| Nonsensical measures | ✓ | ✗ | NA | ✓ | ✗ | NA | ✓ | ✗ | NA |
| | **Mechanical ileus** | | | **COVID-19/ wound infection** | | | **Sentinel bleeding** | | |
| | GPT-3 | GPT-4 | Gemini | GPT-3 | GPT-4 | Gemini | GPT-3 | GPT-4 | Gemini |
| Redundant measures | ✗ | ✗ | NA | ✗ | ✗ | NA | ✓ | ✗ | NA |
| Dangerous measures | ✓ | ✗ | NA | ✗ | ✗ | NA | ✗ | ✗ | NA |
| Inconsistent measures | ✗ | ✗ | NA | ✗ | ✗ | NA | ✗ | ✗ | NA |
| Absence of important measures | ✓ | ✓ | NA | ✓ | ✓ | NA | ✗ | ✗ | NA |
| Nonsensical measures | ✓ | ✗ | NA | ✗ | ✗ | NA | ✗ | ✗ | NA |

LLMs, GPT-3, GPT-4, and Gemini, we intended to showcase the distinct capabilities and sources of error that occur across generations of LLMs for illustrating both their translational potential and respective caveats in surgical patient care.

Our study showed that LLMs like GPT-3, GPT-4, and Gemini demonstrate heterogeneous applicability for clinical decision making. With regard to rather simple tasks such as triaging patients as well as identifying complications and complication mechanisms, GPT-3 and GPT-4 demonstrated comparable results compared to human caregivers but rather overestimated the urgency of the postoperative patient cases. The triage errors that GPT-3 and GPT-4 made were all related to cases that, depending on the clinical presentation, can vary in urgency (mechanical ileus, sentinel bleeding, COVID-19). For example, acute, mechanical ileus in the small intestine often requires immediate intervention. Our case presented a mild form of mechanical ileus in a currently stable patient, with the cause of the ileus symptoms being a tight ostomy. There is currently no clear definition of an emergency situation in an inpatient setting. The performance differences between LLMs and human caregivers showcase that LLMs tend to overestimate the urgency of care situations as compared to humans with clinical experience and expertise. This overestimation of urgency concerning the triaging of patients and identifying complications is, overall, in line with previous findings from other medical fields. [2–4,17–20,29,30] While this overestimation might reduce missing emergencies and increase patient safety, it poses a further risk of constraining physicians and resources. Specifically, overestimating a patient's condition as an emergency limits physicians' abilities to allocate treatment appropriately, decreasing their time for severely ill patients.

A recent study by Palenzuela *et al.* investigated the accuracy of GPT-4 in surgical decision-making compared with surgeons based on five clinical scenarios. This study showed that GPT-4 performed superior to junior residents and equivalent to senior residents and attendings when faced with surgical patient scenarios, showcasing the potential of LLMs as an educational resource for junior residents. While Palenzuela *et al.* used general surgical patient cases and intended to elucidate general diagnoses and treatment paths, we specifically utilized postoperative complication cases and investigated triaging, classification as well as treatment suggestions for these complications. In addition, the study conducted by Palenzuela *et al.* only tested GPT-4, limiting the generalizability of the study. Nevertheless, this study supports our findings and adds further evidence to our study. [4]

To investigate the clinical potential of LLMs in clinical decision making, we evaluated diagnostic and therapeutic measures suggested by different LLMs for management of the six postoperative complication cases. While Gemini refused to provide management suggestions, GPT-3 and GPT-4 provided substantial measures and argued meaningfully. Similar to

the triage performance, GPT-4 surpassed GPT-3 in terms of correctness and completeness of components of the clinical management. A similar performance improvement throughout the development of GPT-type models was also shown in other studies. [29,31,32] Major improvements in the development of GPT-3 to GPT-4 stem from the increased parameters from 175 billion to 1.8 trillion and the ability to process multimodal input. Several additional limitations of LLMs in the context of clinical patient care have been identified in previous studies. For example, Hager *et al.* observed that current LLMs do not accurately diagnose pathologies, follow guidelines, or interpret lab results in the context of intensive care, posing health risks to patients. Additionally, LLMs struggle with integrating into clinical workflows due to issues with instruction adherence and sensitivity to information quantity and order. The clinical implementation of LLMs is often challenged by their integration into hospitals' electronic health records and patient management systems, requiring specific adaptations and refinements. Additionally, two further issues are anticipated regarding this integration. Firstly, intensive user training will be necessary to ensure that it does not create additional burdens in daily work but is perceived as a value-added tool by users. To achieve this, it would be beneficial to include LLMs as part of medical curricula and professional education to educate medical professionals about potential benefits and risks associated with their clinical deployment. Second, sharing highly sensitive healthcare data, even if anonymized, with LLMs, can lead to data privacy concerns that must be adequately addressed. Patients should be informed about the risks and benefits associated with the use of LLMs, and they should have the option to object to the use of LLMs concerning their health data. Overall, our own and others' results indicate that LLMs are not yet suitable for autonomous clinical decision-making. [3] Similarly, Salihu *et al.* found that LLMs can create misunderstandings due to a lack of real-time knowledge updates and contextual sensitivity. Therefore, LLM users need to understand the limitations of LLMs to guarantee quality patient care while LLMs evolve and will become a part of routine patient care. [2]

## Limitations

The limitations of this study originate from the selection of evaluated LLMs and its preclinical stage. First, GPT-3, GPT-4, and Gemini were assessed in this work, and during the study period, several new LLMs have been released, including PaLM, Llama or Vicuna. [33] To enhance LLM performance and applicability in medicine, several improvements could be considered. Integrating more advanced models, like GPT-4.5, may improve accuracy and reliability in clinical settings. Second, our study setting was limited to a retrospective evaluation on retrospective case descriptions. To evaluate LLMs performance in a real-world setting, prospective evaluation is critical. Although our analysis shows the potential of LLMs in postoperative care, their integration into clinical workflows is complex, requiring adaptation to existing systems and ensuring patient data privacy. Our exploratory assessment of counterfactual changes was limited to one patient case, prompted once using GPT-4o, limiting the generalizability of our findings. More extensive testing across various use cases and foundation models, in conjunction with an analysis of changes in human caregivers' responses in reaction to such changes, will be necessary to fully characterize LLM capabilities and reasoning processes for the interpretation of postoperative patient scenarios. Additionally, the study's focus on six postoperative cases may limit the generalizability of its findings, even though these cases represent common and serious complications in general surgery. The use of open-source models could facilitate collaborative development, allowing for extensive validation and customization to specific medical domains. [22] In our study, we deployed a simplified, zero-shot prompting approach to investigate baseline LLM capabilities. We acknowledge that this does not leverage the full potential of LLMs, which achieve better performance with structured and more elaborate prompts. Additionally, combining LLMs with retrieval-augmented generation (RAG) techniques could enhance contextual understanding and accuracy by leveraging external knowledge bases. [21]

## Conclusion

Our findings showcase the potential of newer-generation LLMs like GPT-4 in medical decision-making, clinical quality control, and as an educational resource for less experienced caregivers in the context of postoperative complications. A successful integration of LLMs into clinical practice could help ensure consistently high quality of care in postoperative settings.

## Supporting information

**S1 Table. Postoperative patient cases.**
(DOCX)

**S2 Table. Ideal diagnostic and therapeutic pathways for all cases of postoperative complications according to German and European treatment guidelines as well as current scientific evidence (Wente et al., 2009; Bassi et al., 2017; Kulu et al., 2013; Welsch et al., 2016).**
(DOCX)

**S3 Table. Evaluation of the suspected diagnosis by suspected affected organ system, suspected complication mechanism, and suspected exact complication across all postoperative cases, disaggregated by competency levels of human caregivers.** The table outlines the number of correct triage assessments. Specification of the organ system was mandatory, while specification of complication mechanism and exact complication were voluntary. Data were published previously in Schwarzkopf et al., JMIR Serious Games (2023).
(DOCX)

**S4 Table. Evaluation of counterfactual changes in patient parameters and LLM temperature using GPT-4o for emergency assessment and identification of underlying complications in the anastomotic leakage case.**
(DOCX)

**S5 Table. Evaluation of counterfactual changes in patient parameters and LLM temperature using GPT-4o for suggesting diagnostic and therapeutic measures in the anastomotic leakage case.**
(DOCX)

## Author contributions

**Conceptualization:** Sophie-Caroline Schwarzkopf, Fiona Kolbinger.

**Data curation:** Sophie-Caroline Schwarzkopf, Jean-Paul Bereuter, Mark Enrik Geissler, Fiona Kolbinger.

**Formal analysis:** Sophie-Caroline Schwarzkopf, Jean-Paul Bereuter, Mark Enrik Geissler, Fiona Kolbinger.

**Funding acquisition:** Jürgen Weitz, Marius Distler, Fiona Kolbinger.

**Investigation:** Sophie-Caroline Schwarzkopf, Jean-Paul Bereuter, Mark Enrik Geissler, Fiona Kolbinger.

**Methodology:** Sophie-Caroline Schwarzkopf, Jean-Paul Bereuter, Mark Enrik Geissler, Fiona Kolbinger.

**Project administration:** Fiona Kolbinger.

**Resources:** Jürgen Weitz, Marius Distler, Fiona Kolbinger.

**Supervision:** Jürgen Weitz, Marius Distler, Fiona Kolbinger.

**Validation:** Sophie-Caroline Schwarzkopf, Jean-Paul Bereuter, Mark Enrik Geissler, Fiona Kolbinger.

**Visualization:** Sophie-Caroline Schwarzkopf, Jean-Paul Bereuter, Mark Enrik Geissler, Fiona Kolbinger.

**Writing – original draft:** Sophie-Caroline Schwarzkopf, Jean-Paul Bereuter, Mark Enrik Geissler, Fiona Kolbinger.

**Writing – review & editing:** Jürgen Weitz, Marius Distler.

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
