## [Decision Letter · Decision Letter 0]

PDIG-D-24-00550Postoperative Complication Management: How Do Large Language Models Measure Up to Human Expertise?PLOS Digital Health Dear Dr. Kolbinger, Thank you for submitting your manuscript to PLOS Digital Health. After careful consideration, we feel that it has merit but does not fully meet PLOS Digital Health's publication criteria as it currently stands. Therefore, we invite you to submit a revised version of the manuscript that addresses the points raised during the review process. Please pay particular attention to the comments of Reviewer #3 and ensure that your research meets the highest reproducibility standards (e.g. comment about prompting strategy). Please submit your revised manuscript within 60 days Apr 26 2025 11:59PM. If you will need more time than this to complete your revisions, please reply to this message or contact the journal office at digitalhealth@plos.org.  Please include the following items when submitting your revised manuscript:* A rebuttal letter that responds to each point raised by the editor and reviewer(s). You should upload this letter as a separate file labeled '?>Response to ReviewersRevised Manuscript with Track ChangesManuscript**Additional Editor Comments (if provided):****Reviewers' Comments:**

**Comments to the Author**

1. Does this manuscript meet PLOS Digital Health’s publication criteria?

Reviewer #1: Yes

Reviewer #2: Yes

Reviewer #3: Yes

2. Has the statistical analysis been performed appropriately and rigorously?

Reviewer #1: Yes

Reviewer #2: Yes

Reviewer #3: Yes

3. Have the authors made all data underlying the findings in their manuscript fully available (please refer to the Data Availability Statement at the start of the manuscript PDF file)?

Reviewer #1: Yes

Reviewer #2: No

Reviewer #3: No

4. Is the manuscript presented in an intelligible fashion and written in standard English?

Reviewer #1: Yes

Reviewer #2: Yes

Reviewer #3: No

Reviewer #1: The manuscript addresses a timely and significant topic by assessing the role of large language models (LLMs) in postoperative complication management, an area with substantial potential for clinical impact. The inclusion of multiple LLMs (GPT-3, GPT-4, and Gemini) and comparisons with human caregiver performance provides a thorough evaluation of their capabilities and limitations. The study employs a well-structured methodology using realistic cases to systematically assess triage, diagnosis, and management, enhancing its rigor and clarity. Furthermore, the adherence to ethical standards, including IRB approval and participant consent, adds credibility and robustness to the research.

Despite its strengths, the paper has room for improvement. While it highlights the overestimation of urgency by GPT models, a deeper discussion on how this might impact clinical workflows would enhance the implications section. For example, could this bias result in unnecessary resource allocation or potentially improve patient safety by erring on the side of caution? Additionally, while the paper notes advancements in GPT models, a concise analysis of the factors driving these improvements, such as enhanced training data, architectural refinements, or more detailed prompt engineering, would provide valuable context. Lastly, the discussion acknowledges the need for prospective evaluation but could further elaborate on practical challenges in integrating LLMs into real-world clinical settings, including issues related to model deployment, clinician training, and data privacy.

Reviewer #2: This manuscript is well-written, and evaluates the performance of 3 LLMS in evaluating postop complications. A couple minor comments and suggestions:

1) The competency level of the "human caregivers" are very likely to be heterogenous, given that it spanned medical students to board certified surgeons. Would it be possible to conduct disaggregated analysis comparing the performance of the different subgroups against the LLMs? This might inform us as to whether LLMs are most useful to people at a certain level of training.

Moreover, it strikes me that the difficulty of the 6 cases are different (for example, stroke was very straightforward, but anastomotic leak with the strong guarding was not). Therefore, this reviewer suggests including in Table 1 what kind of human caregivers answered which of the cases to better provide context to the results.

In addition, please further describe your human cohort. Please define "surgical experts". Please describe whether the human caregivers were all from a certain university (if so, this should be mentioned in the limitations), what sampling design was used, and whether there was any power analysis done.

2) Is there a reason anastomotic leakage was the case presented to human caregivers much more than others?

3) Please clarify line 225-227. In comparison to human caregivers, we observed significant performance differences for the

tested LLMs (humans: 89.8% correctly rated vs. GPT-3: 97.7% correctly rated vs. GPT-4:

100% correctly rated vs. Gemini: 51.7%, p < 0.01).  clarify in what outcome this was? which among the comparisons was p <0.01?

This is confusing because the next sentence seems to contradict it. : While GPT-3 and GPT-4 were not

228 significantly different from human caregivers (humans: 89.8% correctly rated vs. GPT-3:

229 97.7% correctly rated vs. GPT-4: 100% correctly rated, p = 0.34), Gemini showed a

230 significantly inferior performance (humans: 89.8% correctly rated vs. Gemini: 51.7%, p < 0.01).

4) Finally, please provide additional details on how the reviewers evaluated the responses to the cases. For instance, in table 1 for COVID 19 GPT-4 got a score of 3/10-- what does this entail?

Reviewer #3: This article compares three language models to a pool of human experts on the tasks of postoperative complication management.

Methodological comments:

To better judge the validity and generality of findings, it would have been helpful to learn more about the exact prompting strategies that were tried out here. Some examples that I suspect the authors may have employed, but were not described are few-shot demonstrations, CoT, templating and the likes.

Similarly, I am curious how exactly the guidelines were provided to the models. I could imagine fine-tuning, RAG, knowledge bases, soft prompts, or memory-augmented networks but am left to speculate. More clarity on this aspect of the study would be helpful.

More information on the simulated patient scenario would be interesting. In particular, how long post-op are these artificial patients supposed to be? If they are recent ops, they will likely have rich sensor information available (e.g., heart rates, blood pressures, hepatology, etc.) that would be very effective inputs to the model beyond mere text.

More than stability across re-runs (which can be easily fixed via temperature etc.) I would have been interested to see stability across trivial counterfactual modifications (age one year older, temperature 0.1 degree higher, etc.) of the case to assess whether functionally similar patients receive similar suggestions.

Minor comment:

The list of abbreviations includes “serious games” which seems unrelated from the content of the paper and might be a copy/paste artifact from the same authors’ previous submission [23].

**Do you want your identity to be public for this peer review?** For information about this choice, including consent withdrawal, please see our Privacy Policy

Reviewer #1: No

Reviewer #2: No

Reviewer #3: No

**Figure resubmission:****Reproducibility:** To enhance the reproducibility of your results, we recommend that authors of applicable studies deposit laboratory protocols in protocols.io, where a protocol can be assigned its own identifier (DOI) such that it can be cited independently in the future. Additionally, PLOS ONE offers an option to publish peer-reviewed clinical study protocols. Read more information on sharing protocols at https://plos.org/protocols?utm_medium=editorial-email&utm_source=authorletters&utm_campaign=protocols

---

## [Decision Letter · Decision Letter 1]

PDIG-D-24-00550R1Postoperative Complication Management: How Do Large Language Models Measure Up to Human Expertise?PLOS Digital Health Dear Dr. Kolbinger, Thank you for submitting your manuscript to PLOS Digital Health. After careful consideration, we feel that it has merit but does not fully meet PLOS Digital Health's publication criteria as it currently stands. Therefore, we invite you to submit a revised version of the manuscript that addresses the points raised during the review process. Please submit your revised manuscript within 30 days Jul 11 2025 11:59PM. If you will need more time than this to complete your revisions, please reply to this message or contact the journal office at digitalhealth@plos.org.  Please include the following items when submitting your revised manuscript:* A rebuttal letter that responds to each point raised by the editor and reviewer(s). You should upload this letter as a separate file labeled '?>Response to ReviewersRevised Manuscript with Track ChangesManuscript**Journal Requirements:****Additional Editor Comments (if provided):****Reviewers' Comments:**

**Comments to the Author**

Reviewer #3: All comments have been addressed

publication criteria?

Reviewer #3: Partly

3. Has the statistical analysis been performed appropriately and rigorously?

Reviewer #3: Yes

4. Have the authors made all data underlying the findings in their manuscript fully available (please refer to the Data Availability Statement at the start of the manuscript PDF file)?

Reviewer #3: Yes

5. Is the manuscript presented in an intelligible fashion and written in standard English?

Reviewer #3: Yes

Reviewer #3: I would like to thank the authors for their clarifications. A purely zero-shot prompting approach without any form of role or system prompt (e.g., the commonly used "you are a helpful assistant") is rather naive and not reflective of the true fidelity these models can reach if used correctly. What is reported here can only be considered a lower bound to model performance. I suggest adding this as a limitation to the study.

Thanks for clarifying that no RAG etc. was used. From the answer, I guess that the guidelines were not made available to the model, and that, instead, we are testing to which degree guideline-conforming behavior is captured by the models' internal parameters.

The counterfactual analysis is interesting. I might have been cryptic in my comment when talking about temperatures in two contexts. What I meant to say was that LM temperature can be used in order to make model behavior consistent across repetitions of the same input. When giving examples of case counterfactuals, however, I was thinking of the patient's body temperature as a potential biomarker to change. The LM temperature parameter should not be used as a counterfactual.

**Do you want your identity to be public for this peer review?** For information about this choice, including consent withdrawal, please see our Privacy Policy

Reviewer #3: No

**Figure resubmission:****Reproducibility:** To enhance the reproducibility of your results, we recommend that authors of applicable studies deposit laboratory protocols in protocols.io, where a protocol can be assigned its own identifier (DOI) such that it can be cited independently in the future. Additionally, PLOS ONE offers an option to publish peer-reviewed clinical study protocols. Read more information on sharing protocols at https://plos.org/protocols?utm_medium=editorial-email&utm_source=authorletters&utm_campaign=protocols

---

## [Editor Report · Decision Letter 2]

Postoperative Complication Management: How Do Large Language Models Measure Up to Human Expertise?

PDIG-D-24-00550R2

Dear Dr. Kolbinger,

We are pleased to inform you that your manuscript 'Postoperative Complication Management: How Do Large Language Models Measure Up to Human Expertise?' has been provisionally accepted for publication in PLOS Digital Health.

Best regards,

Philipp Berens

Academic Editor

PLOS Digital Health